# Prevalence of Individuals at Clinical High-Risk of Psychosis in the General Population and Clinical Samples: Systematic Review and Meta-Analysis

**DOI:** 10.3390/brainsci11111544

**Published:** 2021-11-20

**Authors:** Gonzalo Salazar de Pablo, Scott W. Woods, Georgia Drymonitou, Héctor de Diego, Paolo Fusar-Poli

**Affiliations:** 1Early Psychosis: Interventions and Clinical-detection (EPIC) Lab, Department of Psychosis Studies, Institute of Psychiatry, Psychology & Neuroscience, King’s College London, London SE5 8AB, UK; gonzalo.salazar_de_pablo@kcl.ac.uk; 2Institute of Psychiatry and Mental Health, Department of Child and Adolescent Psychiatry, Hospital General Universitario Gregorio Marañón School of Medicine, Universidad Complutense, Instituto de Investigación Sanitaria Gregorio Marañón (IiSGM), CIBERSAM, 28040 Madrid, Spain; hectordediego@salud.madrid.org; 3Department of Child and Adolescent Psychiatry, Institute of Psychiatry, Psychology & Neuroscience, King’s College London, London WC2R 2LS, UK; 4Child and Adolescent Mental Health Services, South London and Maudsley NHS Foundation Trust, London SE5 8AZ, UK; 5Department of Psychiatry, Yale University School of Medicine, New Haven, CT 06520, USA; scott.woods@yale.edu; 6Sussex Partnership NHS Foundation Trust, Basingstoke RG21 8UN, UK; georgia.drymonitou@nhs.net; 7OASIS Service, South London and Maudsley NHS Foundation Trust, London SE5 8AZ, UK; 8Department of Brain and Behavioral Sciences, University of Pavia, 27100 Pavia, Italy; 9National Institute for Health Research, Maudsley Biomedical Research Centre, South London and Maudsley NHS Foundation Trust, London SE5 8AZ, UK

**Keywords:** schizophrenia, CHR, prevention, clinical settings, community, meta-analytic evidence, clinical high-risk of psychosis

## Abstract

(1) The consistency and magnitude of the prevalence of Clinical High-Risk for Psychosis (CHR-P) individuals are undetermined, limiting efficient detection of cases. We aimed to evaluate the prevalence of CHR-P individuals systematically assessed in the general population or clinical samples. (2) PRISMA/MOOSE-compliant (PROSPERO: CRD42020168672) meta-analysis of multiple databases until 21/01/21: a random-effects model meta-analysis, heterogeneity analysis, publication bias and quality assessment, sensitivity analysis—according to the gold-standard CHR-P and pre-screening instruments—leave-one-study-out analyses, and meta-regressions were conducted. (3) 35 studies were included, with 37,135 individuals tested and 1554 CHR-P individuals identified (median age = 19.3 years, Interquartile range (IQR) = 15.8–22.1; 52.2% females, IQR = 38.7–64.4). In the general population (k = 13, *n* = 26,835 individuals evaluated), the prevalence of the CHR-P state was 1.7% (95% Confidence Interval (CI) = 1.0–2.9%). In clinical samples (k = 22, *n* = 10,300 individuals evaluated), the prevalence of the CHR-P state was 19.2% (95% CI = 12.9–27.7%). Using a pre-screening instrument was associated with false negatives (5.6%, 95% CI = 2.2–13.3%) and a lower CHR-P prevalence (11.5%, 95% CI = 6.2–20.5%) compared to using CHR-P instruments only (28.5%, 95% CI = 23.0–34.7%, *p* = 0.003). (4) The prevalence of the CHR-P state is low in the general population and ten times higher in clinical samples. The prevalence of CHR-P may increase with a higher proportion of females in the general population and with a younger population in clinical samples. The CHR-P state may be unrecognized in routine clinical practice. These findings can refine detection and preventive strategies.

## 1. Introduction

The clinical high-risk for psychosis (CHR-P) paradigm aims to improve the course of psychotic disorders through prevention in a subgroup of individuals with early symptoms or antecedent features (i.e., indicated prevention) [1,2]. Accordingly, CHR-P individuals are mostly defined by the presence of attenuated psychotic symptoms [3,4], in addition to functional impairment [5] and an accumulation of numerous risk factors for psychosis [6,7]. Because of these problems, such individuals typically display help-seeking behaviours and access specialised CHR-P when available [8].

The characterisation and designation of a CHR-P state are essential to guide the subsequent preventive interventions [2], which have the potential to improve clinical outcomes [8]. Briefly, CHR-P individuals present with subtle attenuated psychotic symptoms lasting on average for more than a year [5], frequently in association with comorbid conditions [3], impairments in cognition, and social cognition [6]. Because of these issues, their quality of life may be impaired [7]. Brain changes have also been observed, although consistent replication and clinical translations are still not fully determined [8].

“CAARMS and SIPS deliver a comparable prevalence of CHR-P cases, likely based on their excellent and comparable psychometric performance to discriminate those at risk or not [7]. Although the SIPS has shown a relatively higher sensitivity for the prediction of psychosis than the CAARMS [9], this difference did not influence the prevalence of cases identified. Attenuated Psychosis Syndrome diagnosis has recently been introduced to DSM-5 and is associated with comparable prognostic accuracy [3]”.

Psychotic disorders are frequent in the general population. The pooled incidence of psychotic disorders is 26·6 per 100,000 person-years [10]. Efficient detection of CHR-P individuals is the key rate-limiting step for preventive pathways to care [2]. Recent studies indicate that most young people at possible risk of psychosis remain undetected until they develop the disorder [11,12]. Quantifying the magnitude of those potentially undetected is limited by the unknown prevalence of CHR-P cases. Detection of CHR-P individuals is largely based on idiosyncratic recruitment strategies, which combine help-seeking behaviors and referrals made on suspicion of a psychosis risk [11]. Therefore, to estimate the prevalence of CHR-P, it is necessary to focus on studies systematically assessing CHR-P cases across clinical samples [13,14]. These studies report conflicting findings [13,15,16,17], and the actual prevalence of CHR-P individuals within clinical samples is not determined. Some studies have also systematically investigated the prevalence of CHR-P in the general population [18,19,20], despite the CHR-P paradigm [2] not being primarily conceived to prevent psychosis in healthy individuals (universal prevention). The findings are also conflicting [21,22,23]: a previous systematic review found that the prevalence of the CHR-P state ranged from 1–8% in educational settings [24]. A further knowledge gap relates to the impact of using pre-screening instruments before the CHR-P evaluation to enhance detection [25,26]. Finally, while the prevalence of CHR-P cases may be moderated by age [27], gender [1], or geographical location [28], the actual impact of these factors is undetermined.

No systematic reviews and meta-analyses have addressed these gaps of knowledge. Our primary aim was to systematically review the consistency and magnitude of the prevalence of CHR-P individuals in the general population and clinical samples. Our secondary aim was to address the impact of using pre-screening strategies, as well as to test factors that may moderate the prevalence of CHR-P. The overarching goal of this study is to provide novel knowledge on the prevalence of the CHR-P state to facilitate the detection of CHR-P individuals.

## 2. Materials and Methods

This study (study protocol: PROSPERO CRD42020168672) was conducted in accordance with the “Preferred Reporting Items for Systematic reviews and Meta-Analyses” (PRISMA [29]) (Appendix A) and the “Meta-analysis Of Observational Studies in Epidemiology” (MOOSE [30]) (Appendix A) checklists.

### 2.1. Searches

A systematic search strategy was used to identify relevant articles, and a two-step literature search was implemented by two independent researchers (GSP& HD). The Web of Science database (Clarivate Analytics, which incorporates the Web of Science Core Collection, BIOSIS Citation Index, KCI-Korean Journal Database, MEDLINE, Russian Science Citation Index, and SciELO Citation Index), Cochrane Central Register of Reviews, and Ovid/PsychINFO databases were searched from inception until 21/01/21, in English. The following search terms were used: “risk” OR “prodrom *” OR “ultra-high risk” OR “clinical high risk” OR “attenuat *” OR “high risk” OR “genetic high risk” OR “at risk mental state” OR “at-risk mental state” OR “risk of progression” OR “progression to first-episode” OR “first episode” OR “schizophrenia” OR “schizoaffective disorder” OR “schizophreniform disorder” OR “basic symptoms” AND “Comprehensive Assessment of At-Risk Mental States” OR “CAARMS” OR “Structured Interview for Psychosis-risk Syndromes” OR “SIPS” OR “Bonn Scale for the Assessment of Basic Symptoms” OR “BSABS” OR “Basel Screening Instrument for Psychosis” OR “BSIP” OR “Schizophrenia Proneness Instrument” OR “SPI” OR “SPI-A” OR “SPI-CY” OR “Early Recognition Inventory” OR “ERIraos” OR “Epidemiolog *” AND “psychosis”. Articles identified were screened as abstracts, and the full texts of the eligible articles were assessed against the inclusion/exclusion criteria. We completed our search by looking at medRxiv and PsyArXiv preprint databases and manually reviewing the references of previously published articles before extracting any additional relevant titles. We finally looked for grey literature in the Open Grey database. Whenever relevant meta-analytic data were missing, authors were contacted and invited to share further data.

### 2.2. Inclusion and Exclusion Criteria

Inclusion criteria were: (a) original individual studies; (b) conducted in the general population or in clinical samples (previously operationalised) [31] to include general medical practitioners, community or inpatient adult mental health services, child and adolescent community or inpatient mental health services, early intervention for psychosis services, forensic samples and schools/colleges mental health counselling centres, accident and emergency departments, and general physical health services; (c) providing the prevalence (proportion %) of individuals meeting CHR-P criteria across the individuals assessed according to gold-standard CHR-P psychometric instruments defining the Ultra High Risk (UHR) or Basic Symptoms (BS) state (more details can be found in Methods S1); (d) performing a systematic assessment of all individuals for a putative CHR-P syndrome, operationalised as (i) random sampling, (ii) systematic pre-screening (detailed in Methods S2) followed by gold-standard CHR-P assessment for those above the pre-screening threshold, or (iii) systematic and consecutive CHR-P assessment of all individuals referred to a specific mental health service within a certain period of time using established psychometric instruments (Methods S1); (e) with a minimum quality rating of 50% (i.e., 4/8) according to the modified Newcastle-Ottawa Scale (NOS); (f) published in English. Exclusion criteria were: (a) reviews, clinical cases, or conference proceedings; (b) studies not using established psychometric instruments to detect CHR-P individuals; (c) studies in which individuals were unsystematically assessed for CHR-P (e.g., assessing only those suspected of being at CHR-P or subgroups of individuals referred for a CHR-P assessment and subsequent preventive intervention); (d) overlapping studies. Overlap was determined by looking at the program, the recruitment period, the instruments, and the country in which the study was carried out. The most representative and largest sample was selected in the case of overlapping samples. Because the CHR-P paradigm has been largely developed in individuals referred on suspicion of psychosis-risk, and not systematically assessed for a potential CHR-P state, a large number of studies focusing on CHR-P clinical services were expected to meet the exclusion criterion c.

### 2.3. Data Extraction

Two researchers (GSP & GD) independently extracted data from all the included studies into a Microsoft Excel spreadsheet. Discrepancies were resolved through consensus under the supervision of a third author (PF-P). The variables in the study included the following information: author and year; country; population (general population vs. clinical samples); population type (general or clinical) and age (for the samples assessed and CHR-P individuals); assessment type—pre-screening tool (if applicable) and gold-standard CHR-P assessment instrument; sample size (samples assessed and CHR-P individuals); CHR-P subgroups (proportion of Attenuated Psychotic Symptoms (APS), Brief and Limited Intermittent Psychotic Symptoms (BLIPS/BIPS), Genetic Risk and Deterioration syndrome (GRD), and basic symptoms (BS); sex (% female CHR-P); and quality assessment (see below).

### 2.4. Strategy for Data Synthesis and Effect Measures

We estimated the prevalence of CHR-P individuals systematically assessed in the general population and clinical samples as proportions (95% CI). We used a random-effects model, as heterogeneity was expected to be high in the studies included. Heterogeneity among study point estimates was assessed using the Q statistic and the I^2^ index. The presence of publication bias in the results was assessed by Egger’s test [32] and by visually inspecting the funnel plots, which are provided in the Appendix A. The use of the “trim and fill” method was planned to correct the effects of any publication bias detected.

We carried out a sensitivity analysis to compare: (a) type of CHR-P interview—studies using the SIPS vs. studies using the CAARMS; and (b) type of assessment—studies using only the SIPS/CAARMS vs. those using first a pre-screening instrument and then the gold-standard CHR-P instrument for individuals testing positive at the pre-screening test. Other sensitivity analyses compared: (c) studies conducted in school/colleges vs. other studies within the general population group; and (d) forensic samples vs. other samples within the clinical samples group. For comprehensiveness, we further evaluated the meta-analytical prevalence of CHR-P individuals testing negative on the pre-screening instruments but later testing positive on the CHR-P assessment (i.e., “false negatives” at the pre-screening assessment). Additional leave-one-out sensitivity analyses evaluated the stability of the meta-analytic findings when each study was removed at a time. We used meta-regression techniques to evaluate the impact of the following predictors of interest: (a) age; (b) sex; (c) quality of the study; and (d) study continent. We used two-sided statistical tests and a significance level of alpha = 0.05. The analyses were carried out with the Meta and Metaprop packages of Stata statistical software version 16 (StataCorp) [33] and the Comprehensive Meta-Analysis Version 3 software [34].

### 2.5. Risk of Bias (Quality) Assessment

The risk of bias was assessed using a modified version of the NOS for cohort and cross-sectional studies, in line with previous studies [1,3,27]. Studies were awarded a maximum of eight points on items related to the representativeness, sample size, group definition, validity and outcomes for cross-sectional studies and representativeness, exposure, outcomes, follow-up period, and loss to follow-up for cohort studies (Appendix A). A minimum quality rating of 50% (i.e., 4/8) was established, and studies with a lower rating were excluded.

## 3. Results

The literature search yielded 8568 citations after we removed duplicated citations, and these were screened for eligibility. The number of full-text articles evaluated for eligibility was 280, with 245 excluded in this step. In total, 35 studies were finally included (Figure 1, Appendix A). The total sample size of the studies was 37,135 (26,835 in the general population and 10,300 in clinical samples). The most frequently used CHR-P instrument was SIPS (65.7%), followed by CAARMS (40.0%) (not mutually exclusive). The most frequently used pre-screening instruments were the Prodromal Questionnaire, different versions (61.1%), and the Prime Screen, different versions (22.2%). The total number of CHR-P individuals identified was 1554 (352 in the general population and 1202 in clinical samples), with a median age of 19.3 years (IQR = 15.8–22.1) and a median % of females of 52.2 (IQR = 38.7–64.4).

### 3.1. Prevalence of the CHR-P State in the General Population

Thirteen independent samples, including 26,835 individuals, provided meta-analytical data to test the prevalence of CHR-P individuals in the general population (Appendix A). Of them, eight studies (61.5%) used the SIPS as their primary detection instrument, while five studies (38.5%) used the CAARMS. According to random-effects meta-analysis, 1.7% of individuals (95% CI = 1.0–2.9%) at CHR-P were found in the general population (Figure 2). The samples studied were mostly restricted to adolescents and young adults, with the mean age of samples ranging from 13.9 to 24.0 years (Appendix A).

There was no significant difference in the prevalence of the CHR-P state between studies using the SIPS, prevalence 2% (k = 8, *n* = 18,881 total, 197 CHR-P, 95% CI = 0.9–4.5%) and studies using the CAARMS, prevalence 1.5% (k = 5, *n* = 7954 total, 138 CHR-P, 95% CI = 0.6–3.0%) (*p* = 0.469). There was also no significant difference between studies using only the established CHR-P instrument, prevalence 2.7% (k = 4, *n* = 6613 total, 122 CHR-P, 95% CI = 1.1–6.7%) and those using a two-step approach with the pre-screening instrument first followed by an established CHR-P instrument, prevalence 1.5% (k = 9, *n* = 20,222 total, 213 CHR-P, 95% CI = 0.8–3.0%) (*p* = 0.321). There were no significant differences between studies conducted in school/colleges (prevalence: 1.5% (k = 6, *n* = 8109 total, 88 CHR-P, 95% CI = 0.6–3.3%) and other general population studies (prevalence: 1.9%, k = 7, *n* = 18,726 total, 247 CHR-P, 95% CI = 0.9–4.0%, *p* = 0.684). There was only one study assessing with CHR-P instruments those individuals who tested negative on pre-screening assessments in the general population [35]. There were no publication biases (Egger’s test Intercept = -1.833, t = 0.690, *p* = 0.504) (see funnel plot in Appendix A).

### 3.2. Prevalence of the CHR-P State in Clinical Samples

Twenty-two independent samples, including 10,300 individuals, provided meta-analytical data to test the prevalence of CHR-P individuals in clinical samples (Appendix A). Of them, 15 studies (68.2%) used the SIPS as their primary detection instrument, while seven studies (31.8%) used the CAARMS. The prevalence of the CHR-P state in clinical samples was 19.2% (95% CI = 12.9–27.7%) (Figure 3. The clinical samples studied were also fundamentally restricted to adolescents and young adults, with the mean age of samples ranging from 15.3 to 27.9 years (Appendix A).

There were no significant differences between studies using the SIPS, prevalence 22.6% (k = 15, *n* = 4429 total, 634 CHR-P, 22.1% 95% CI = 15.0–31.4%) and those using the CAARMS, prevalence 14.6% (k = 7, *n* = 5871 total, 585 CHR-P, 95% CI = 5.7–32.6%) (*p* = 0.385). However, studies using only the established CHR-P instruments detected a higher prevalence of 28.5% (k = 13, *n* = 2104 total, 632 CHR-P, 95% CI = 23.0–34.7%) compared to those studies employing a two-step approach with the pre-screening instrument first followed by an established CHR-P instrument, prevalence 11.5% (k = 9, *n* = 8196 total, 587 CHR-P, 28.5 95% CI = 6.2–20.5%) (*p* = 0.003). There were no significant differences in forensic samples (prevalence: 11.8%, k = 3, *n* = 973 total, 84 CHR-P, 95% CI = 3.7–31.7%) and other clinical samples (prevalence: 20.7%, k = 19, *n* = 9327 total, 1135 CHR-P, 95% CI = 13.4–30.5%) (*p* = 0.332). The prevalence of false negatives meeting CHR-P criteria after an initial negative pre-screening was 5.6% (95% CI = 2.2–13.3%, k = 4, *n* = 458 total, 34 CHR-P) (see forest plot in Figure 2). There were no publication biases (Egger’s test Intercept = 3.797, t = 1.050, *p* = 0.306) (see funnel plot in Appendix A). Additional leave-one-out sensitivity analyses can be found in Results S1 and Appendix A.

### 3.3. Effect of Moderators

Heterogeneity across general population studies was high (Q = 261.176, I^2^ = 95.405, *p* < 0.001). Heterogeneity across clinical sample studies was also high (Q = 1053.778, I^2^ = 98.077, *p* < 0.001). In the general population, meta-regressions revealed a significant increase in the prevalence of CHR-P individuals with an increasing proportion of females (β = 0.045, *p* = 0.041) (Figure 4A), while there was no effect for age, study quality, and the continent in which the study was carried out (all *p* > 0.05). Within clinical samples, meta-regressions revealed a significant increase in the prevalence of CHR-P individuals with younger ages (β = −0.160, *p* < 0.001) (Figure 4B), while there was no effect for the proportion of females, study quality, and the continent in which the study was carried out (Appendix A) (all *p* > 0.05).

### 3.4. Quality of the Included Studies

Study quality scores ranged from 4 to 8. The overall median quality score for the included studies was 6 (IQR = 5–7) (Appendix A).

## 4. Discussion

This is the first systematic review and meta-analysis to evaluate the prevalence of CHR-P in the general population and clinical epidemiology samples. We have two main findings: (1) the prevalence of CHR-P in the general population of adolescents and young adults was low, at 1.7%; and (2) the prevalence of CHR-P in the population of adolescents and young adults entering mental health services for the first time was more than ten times higher, at 19.2%.

To our best knowledge, this is the largest meta-analysis addressing the prevalence of the CHR-P state in the general population and clinical samples, with a final dataset encompassing 35 studies relating to a total of 37,135 young individuals who were assessed for a potential CHR-P state.

The first main finding is that in the general population, despite some meta-analytic heterogeneity, about 1.7% of young individuals met CHR-P criteria. The low prevalence of interview-based CHR-P designations in the general population contrasts sharply with the reported prevalence of “psychotic-like experiences” [36] measured through self-administered questionnaires [37]. Psychotic-like experiences are relatively frequent at the population level (prevalence about 8% in young adults aged 24 [22]) and poorly predictive of psychosis onset (risk of psychosis: 0.5–1% per year [22]). However, the true “psychotic nature” of these manifestations is questionable [38], and they cannot be conflated with the CHR-P assessment. The CHR-P state requires an experienced and trained clinician to distinguish pathological from non-pathological phenomena [39], and it is not common in the general population (only 1.7% of individuals), being highly predictive of psychosis onset (risk of psychosis: 20% at two years [8,27,40]). Therefore, our 1.7% prevalence should be understood to reflect the occurrence of the categorical and syndromic CHR-P state in the general population and not of continuously distributed transitory psychotic-like experiences of little psychopathological meaning. Overall, the low prevalence of CHR-P in the general population using CHR-P diagnostic instruments supports the validity of the CHR-P construct. Moreover, the low (1.7%) CHR-P prevalence in the general population is consistent with the meta-analytic (median) prevalence of any psychotic disorder (across 73 studies published between 1990 and 2015), which cumulates to 0.389% (12-month prevalence = 0.403%, median lifetime prevalence = 0.749% [41]). The prevalence of psychosis in the general population roughly corresponds to about 22.9% of the CHR-P cases estimated in the general population (0.389/1.7). This estimate approximately corresponds with the most recent estimates of transition to psychosis within CHR-P individuals, of about 20% at two years [40]. These numbers suggest that the magnitude of the prevalence of CHR-P cases in the general population estimated by this study is coherent with the current empirical knowledge-although this comparison is not validated by formal testing and should therefore be interpreted cautiously.

Another population-level finding of this study is that the use of pre-screening instruments in the general population emerges as a good detection approach. This is likely due to the high sensitivity of pre-screening instruments such as the Prodromal Questionnaire-Brief Version (PQ-B) [42] (sensitivity = 89% at ≥3 score cut-off) or the Prime Screen-Revised (PS-R) [15] (sensitivity = 100% at ≥10 score cut-off), which are the most widely employed in the general population. More recently, online pre-screening instruments have also been developed to further help identify CHR-P individuals at a large scale [34]. Using pre-screening instruments in large populations offers competitive logistic advantages compared to conducting extensive CHR-P assessments. Furthermore, there is compelling evidence against intense outreach strategies that systematically employ CHR-P instruments outside clinical samples [7]. These approaches are not recommended [43] because they dilute the level of risk enrichment [44,45] and therefore lead to a clinically meaningless identification of those genuinely developing psychosis [10]. Our findings confirm that community outreach and recruitment of CHR-P cases from the general population should only be considered if adequate risk enrichment strategies can be implemented via pre-screening assessments [10], in particular, via digital strategies [46]: 

The second core finding is that the prevalence of CHR-P cases in the clinical samples was over ten times higher than in the general population: 19.2%. To our best knowledge, this is the first time their consistency and magnitude have been formally appraised at a meta-analytic level. The high prevalence of CHR-P cases in the clinical samples is likely attributable to the higher accumulation of risk factors [1,47] and functional impairments (and, therefore, the higher likelihood of transitioning to psychosis). This finding aligns with the indicated preventive nature of the CHR-P paradigm and confirms that clinical settings play an important role in risk enrichment [44,45]. The latter is modulated by the type of outreach and referral source [48]: it is a higher risk in secondary mental health services [49,50], intermediate in primary care, and lower in the community [29,45]. Risk enrichment accounts for the majority of the transition risk for psychosis eventually observed [43].

There are several empirical implications of this study for clinical practice and research (e.g., recruitment into studies). The main implication is to have provided the first meta-analytic prevalence of the CHR-P state in the general population and clinical samples. The general population estimate can be extremely useful for computing population-level epidemiological measures that quantify the impact of preventive approaches (e.g., the population attributable fraction, PAF). Furthermore, this study demonstrates that, within clinical samples, about five young people should be tested to identify at least one meeting the CHR-P criteria. This picture may be moderated by the use of pre-screening instruments, which led to about 5.6% false negatives, albeit only a few studies reported these data. CHR-P individuals accessing clinical services are typically complex, present with comorbid mental disorders, suicidal ideation, and self-harm, as well as impairments in work/educational functioning, social functioning, and quality of life [5]. Pre-screening instruments may have more difficulties in discriminating between these intertwined mental health problems (Discussion S1).

The clinical assessment of CHR-P states should be carried out within specialised community mental health services or specialised research settings. Policymakers should ensure these services are available for help-seeking individuals across different countries. At the moment, services have only been implemented in a relatively small subset of countries [50].

Therefore, in clinical samples, the use of CHR-P instruments per se, without pre-screening instruments, may be sufficient. The high prevalence of CHR-P “hiding in plain sight” among adolescents and young adults in clinical services suggests that even though individuals are receiving care, their CHR-P symptoms may frequently be going unrecognised in clinics that are not specifically evaluating for them and the opportunity to monitor these individuals and identify transitions to frank psychosis as soon as possible may be being missed. Interestingly, this finding of unrecognised CHR-P cases in clinical services may also explain the putative prognostic value of non-psychotic mental disorders for later schizophrenia [11,12,13]. This raises the possibility that psychosis onset from non-psychotic disorders may be associated with undetected comorbid CHR-P. Lastly, the high prevalence of CHR-P in clinical services may offer an untapped resource for identifying potential research participants. Since obtaining sufficient sample sizes for clinical trials within available resources is one of the principal obstacles to the development of new interventions specifically for CHR-P individuals, the possibility of improved detection of these CHR-P patients could speed the development of needed preventive interventions.

As additional findings, our meta-regressions found that the population-level prevalence of CHR-P state is increased in studies with a higher proportion of females. Affective features, social conflict, and help-seeking behaviours occur more often in females during the phases preceding psychosis onset [51], which may explain these findings. Some authors have previously recommended gender-sensitive refinement of outreach campaigns in this field [52]. At the same time, this finding contrasts with the epidemiological distribution of psychosis risk (at least for schizophrenia-spectrum psychoses), which is typically associated with a predominance in males [1,47]. One possible interpretation is that the relatively higher number of CHR-P females may include more false-positive CHR-P individuals who will eventually not develop psychosis or develop affective—as opposed to schizophrenia-spectrum—psychoses (see limitations). Meta-regression analyses did not uncover any effect for the continent, supporting the global transportability of CHR-P clinical services across different cultures and contextual specifiers [50]. In line with this, we have recently demonstrated that the CHR-P approach has been globally validated in Western Europe (51.1%), North America (17.0%), East Asia (17.0%), Australia (6.4%), South America (6.4%), and even in Africa (2.1%), albeit to a smaller scale [28]. The observed international prevalence of CHR-P features, although awaiting further replication in larger population-level studies, holds clinical value to inform the widescale implementation of specialised services to take care of individuals with emerging mental disorders [50]. Policymakers should make sure these services are available, particularly for help-seeking individuals under clinical care in mental health settings. These services should monitor not only transition to psychosis but also clinically relevant symptoms as negative symptoms, functioning and remission [7,53].

We also found an increased prevalence of CHR-P individuals with younger clinical samples. A recent large-scale meta-analysis of epidemiological studies found that about half of psychotic disorders have an onset before 25 years, with a peak age of onset of 20.5 years [54]. A potential explanation for our findings is that children and adolescents may experience an extended period of risk compared to adult populations [55]. However, it is also likely that the youngsters have a less crystallised and more dynamic, evolving presentation that may be more broadly captured by the current CHR-P tools.

The first limitation is that, although the studies included had systematically assessed individuals for a potential CHR-P state, completion of the CHR-P assessment was based on voluntary consent to research. Furthermore, only a subset of studies using a pre-screening assessment carefully tested for potential false negatives. Second, the CHR-P assessments were typically based on retrospective interviews, which may be prone to recall bias. We limited our study to established psychometric instruments (combined or not with pre-screening instruments), which have shown good reliability and are widely used for the detection of CHR-P individuals. Yet, such individuals may underreport their symptoms at the baseline; an extended assessment is offered in some CHR-P services. Third, despite our comprehensive approach, there remains limited evidence regarding studies conducted in the general population (13 studies). Although this number was sufficient to conduct the meta-analysis, the sensitivity analyses, and the meta-regressions, additional epidemiological research in this field is needed. Fourth, most studies evaluated only the attenuated psychosis symptoms CHR-P subgroup. Fifth, heterogeneity was substantial. We conducted meta-regressions to evaluate the influence of several moderators on our results but were unable to quantify the impact of other potentially meaningful factors, such as the CHR-P subgroups, ethnic, or healthcare system differences. Sixth, as the current findings are essentially based on cross-sectional analyses, additional work is required to determine the extent to which the likelihood of developing psychosis of the CHR-P differs in general and clinical samples, as has been observed in previous studies [29].

## 5. Conclusions

The prevalence of CHR-P cases in the general population is 1.7%, while it is over ten times higher, at 19.2%, in clinical samples. The CHR-P state may be commonly going unrecognized in routine clinical practice. These findings can impact the refinement of detection strategies for these patients and, ultimately, preventive efforts.

## Figures and Tables

**Figure 1 brainsci-11-01544-f001:**
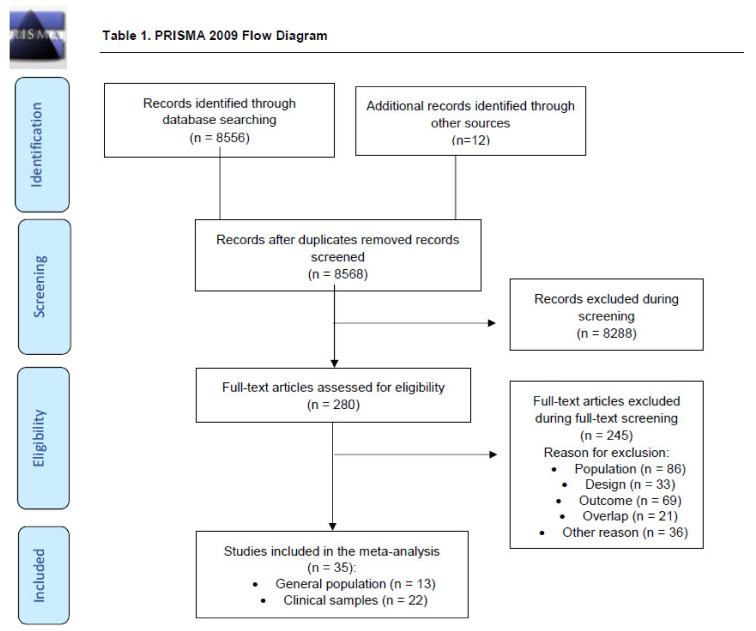
PRISMA 2009 Flow Diagram.

**Figure 2 brainsci-11-01544-f002:**
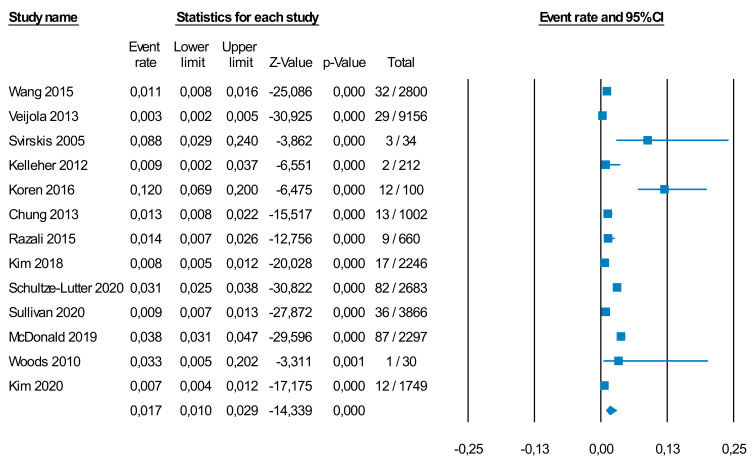
Random effect meta-analysis (forest plot) of the prevalence of CHR-P individuals in the general population. Of the individuals in the general population, 1.7% (95% CI=1.0-2.9%) were found to be at CHR-P.

**Figure 3 brainsci-11-01544-f003:**
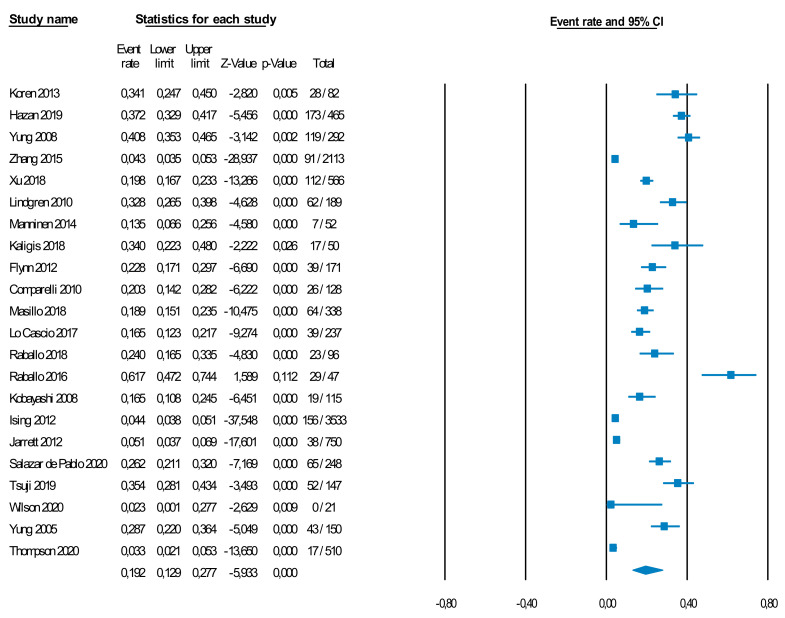
Random effect meta-analysis (forest plot) of the prevalence of CHR-P individuals in clinical samples. A total of 19.2% (95% CI = 12.9–27.7%) of individuals in the clinical samples were found to be at CHR-P.

**Figure 4 brainsci-11-01544-f004:**
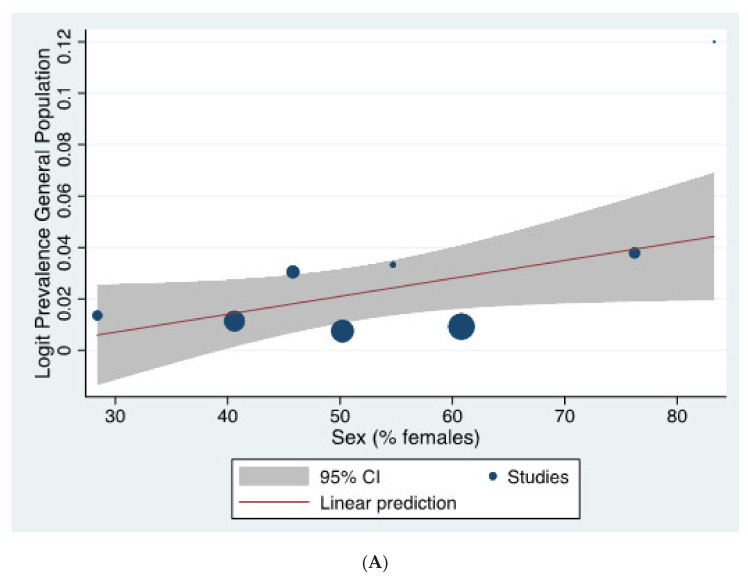
(**A**) Meta-regressions of the influence of sex (% female) on the prevalence of CHR-P in the General Population. A significant increase in the prevalence of CHR-P individuals with an increasing proportion of females (β = 0.045, *p* = 0.041) was found in the general population. (**B**) Meta-regression of the influence of age on the prevalence of CHR-P in clinical samples. A significant decrease in the prevalence of CHR-P individuals with an increasing age (β = −0.160, *p* < 0.001) was found in clinical samples.

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
