# Peer review of "Prevalence of Individuals at Clinical High-Risk of Psychosis in the General Population and Clinical Samples: Systematic Review and Meta-Analysis"

_brainsci, 2021, doi:10.3390/brainsci11111544_

Round 1

Reviewer 1 Report

Overall, this is an interesting meta-analysis and quite a well-written manuscript that has the potential to shed more light on the prevalence of individuals at clinical high-risk of psychosis in the general population and clinical samples. The aims of the study are original. Generally, the introduction and discussion are suitable and wide and new literature was referenced in the appropriate context. The methods and statistical meta-analysis are adequate.

Some points need to be considered:

Keywords:
-    I suggest adding to the keywords: clinical high-risk of psychosis.

Introduction:
-    I suggest adding a small paragraph on the importance of diagnosis of a high state risk of psychosis with reasons. The authors should add information about psychiatric symptoms, cognitive and social cognition impairments, quality of life, and brain changes in that population.

Results:
-    The authors should change the "Table 1" to "Figure 1" as a "PRISMA 2009 Flow Diagram".

Discussion:
-    The authors should add information on the importance of the results of their study for clinical practice to the discussion.
-    The authors should add some recommendations about the assessment of at-risk mental state as a potential implication of the results of their study.

Author Response

Thank you for the useful comments. Please find attached the response to reviewers.

REVIEWER 1

  • Overall, this is an interesting meta-analysis and quite a well-written manuscript that has the potential to shed more light on the prevalence of individuals at clinical high-risk of psychosis in the general population and clinical samples. The aims of the study are original. Generally, the introduction and discussion are suitable and wide and new literature was referenced in the appropriate context. The methods and statistical meta-analysis are adequate.

Some points need to be considered:

RESPONSE:

Thank you for the positive feedback

  • Keywords:
    I suggest adding to the keywords: clinical high-risk of psychosis.

RESPONSE:

We have added the keyword “clinical high-risk of psychosis“

  • Introduction:
    -    I suggest adding a small paragraph on the importance of diagnosis of a high state risk of psychosis with reasons. The authors should add information about psychiatric symptoms, cognitive and social cognition impairments, quality of life, and brain changes in that population.

RESPONSE:

We have expanded these points as follows:

“The characterization and designation of a CHR-P state is essential to guide the subsequent preventive interventions1,2 which have the potential to improve clinical outcomes3,4. Briefly, CHR-P individuals present with subtle attenuated psychotic symptoms lasting on average for more than 1 year5, frequently in association with comorbid conditions6, impairments in cognition and social cognition7. Because of these issues their quality of life may be impaired8. Brain changes have also been observed although consistent replication and clinical translations are still not fully determined5

  • Results:
    -    The authors should change the "Table 1" to "Figure 1" as a "PRISMA 2009 Flow Diagram".

RESPONSE:

We have replaced "Table 1" for "Figure 1"

  • Discussion:
    -    The authors should add information on the importance of the results of their study for clinical practice to the discussion.

RESPONSE:

We have included information on the importance of the results of their study for clinical practice to the discussion:

“Our findings confirm that community outreach and recruitment of CHR-P cases from the general population should only be considered if adequate risk enrichment strategies can be implemented via pre-screening assessment9, in particular via digital strategies10”.

“There are several empirical implications of this study for clinical practice and research (e.g., recruitment into studies). The main implication of this study is to have provided the first meta-analytic prevalence of the CHR-P state in the general population and in clinical samples. The general population estimate can be extremely useful to compute population-level epidemiological measures that can quantify the impact of preventive approaches (e.g. the population attributable fraction, PAF). Furthermore, this study demonstrates that within clinical samples, about five young people should be tested to identify at least one meeting the CHR-P criteria. This picture may be moderated by the use of pre-screening instruments, which led to about 5.6% false negatives, albeit only a few studies reported these data. CHR-P individuals accessing clinical services are typically complex, present with comorbid mental disorders, suicidal ideation and self-harm, as well as impairments in work/educational functioning, social functioning and quality of life5. Pre-screening instruments may have more difficulties in discriminating between these intertwined mental health problems (eDiscussion). Therefore, in clinical samples, the use of CHR-P instruments per se, without pre-screening instruments, may be sufficient. The high prevalence of CHR-P “hiding in plain sight” among adolescents and young adults in clinical services suggests that even though individuals are receiving care, their CHR-P symptoms may frequently be going unrecognised in clinics who are not specifically evaluating for them and that the opportunity to monitor these individuals and identify transitions to frank psychosis as soon as possible may be being missed. Interestingly, this finding of unrecognised CHR-P cases in clinical services may also explain the putative prognostic value of non-psychotic mental disorders for later schizophrenia11-13, raising the possibility that psychosis onset from non-psychotic disorders may be associated with undetected comorbid CHR-P. Lastly, the high prevalence of CHR-P in clinical services may offer an untapped resource for the identification of potential research participants. Since obtaining sufficient sample sizes for clinical trials within available resources is one of the principal obstacles to the development of new interventions specifically for CHR-P individuals, the possibility of improved detection of these CHR-P patients could speed the development of needed preventive interventions.”

Additional implications of the current study are appended below:

“Policy makers should make sure these services are available, particularly for help-seeking individuals under clinical care in mental health settings”

“The observed international prevalence of CHR-P features, although awaiting further replication in larger population-level studies, holds clinical value to inform the widescale implementation of specialised services to take care of individuals with emerging mental disorders14”.

6-    The authors should add some recommendations about the assessment of at-risk mental state as a potential implication of the results of their study.

RESPONSE: We have added these recommendations:

“The clinical assessment of CHR-P states should be carried out within specialised community mental health services or specialized research settings. Policy makers should make sure these services are available for help-seeking individuals across different countries. At the moment these services are only implemented in a relatively small subset of countries14,15.“

Reviewer 2 Report

18 October 2021

Manuscript ID: brainsci-1443994

Type: Review

Title: “Prevalence of Individuals at Clinical High-risk of Psychosis in the General Population and Clinical Samples: Systematic Review and Meta-analysis” by Salazar de Pablo G et al., submitted to Brain Sciences

Dear Authors,

Efficient detection of clinical high-risk for psychosis (CHR-P) state is the most important step for the prevention of psychosis; however, little is known about the prevalence of CHR-P. The authors conducted a meta-analysis of the prevalence of CHR-P individuals in the general population and clinical samples. The study concluded that the prevalence of CHR-P is ten times higher in clinical samples than general population and that CHR-P state may remain unrecognized in routine clinical practice.

Please reconsider the following:

  1. A graphical abstract summarizing the manuscript is highly recommended.
  2. Page 1, Abstract: Please present the background, methods, results, and conclusion proportionally.
  3. Page 2, Introduction: Please briefly present the epidemiology of psychosis, psychosis in psychiatric disorders and comorbidity, importance of CHR-P state detection, methods to assess CHR-P and its variability, CHR-P as an important risk biomarker, among others to lead to the importance of this study.
  4. Page 6, Figure 1: The diamond is missing. The sample sizes are preferably presented in the forest plot. Please present short description in the caption.
  5. Page 7, Figure 2: The sample sizes are preferably presented in the forest plot. Please present short description in the caption.
  6. Pages 8,9, Figure 3: Please present short description in the caption.
  7. Pages 9-12, Discussion: The limitation is well discussed. Please describe potentials in the study, the ultimate goal, research or knowledge needed to achieve, the future research direction, and the biggest challenge in this goal, among others.

The manuscript contains three figures, one table and 56 references. The novelty of the content is fairly good; there will be relatively potential impact of the manuscript in the relevant field of research; the standard of English is good;  the study design and methodology are appropriate. The manuscript carries important value presenting the prevalence of CHR-P and need for fine assessment of CHR-P state for routine clinical practice.

Best regards,

Author Response

Thank you for the useful comments. Please find attached the response to reviewers.

REVIEWER 2

Efficient detection of clinical high-risk for psychosis (CHR-P) state is the most important step for the prevention of psychosis; however, little is known about the prevalence of CHR-P. The authors conducted a meta-analysis of the prevalence of CHR-P individuals in the general population and clinical samples. The study concluded that the prevalence of CHR-P is ten times higher in clinical samples than general population and that CHR-P state may remain unrecognized in routine clinical practice.

Please reconsider the following:

  1. A graphical abstract summarizing the manuscript is highly recommended.

RESPONSE: We have added a graphical abstract:

  1. Page 1, Abstract:Please present the background, methods, results, and conclusion proportionally.

RESPONSE: We have reduced the methods and results section and expanded the introduction and discussion:

Abstract: (1) The consistency and magnitude of the prevalence of Clinical High-risk for psychosis (CHR-P) individuals are undetermined, limiting efficient detection of cases. Our aim was to evaluate the prevalence of CHR-P individuals systematically assessed in the general population or clinical samples (2) PRISMA/MOOSE-compliant (PROSPERO: CRD42020168672) meta-analysis of multiple databases until 21/01/21. Random-effects model meta-analysis, heterogeneity analysis, publication bias and quality assessment, sensitivity analysis—according to the gold-standard CHR-P and pre-screening instruments—leave-one-study-out analyses and meta-regressions were conducted (3) 35 studies were included, with 37,135 individuals tested and 1,554 CHR-P individuals identified (median age=19.3 years, IQR=15.8-22.1; 52.2% females, IQR=38.7-64.4). In the general population (k=13, n=26,835 individuals evaluated), the prevalence of CHR-P state was 1.7% (95% CI=1.0-2.9%). In clinical samples (k=22, n=10,300 individuals evaluated), the prevalence of the CHR-P state was 19.2% (95% CI=12.9-27.7%). Using a pre-screening instrument was associated with false negatives (5.6%, 95% CI=2.2-13.3%) and a lower CHR-P prevalence (11.5%, 95% CI=6.2%-20.5%) compared to using CHR-P instruments only (28.5%, 95% CI=23.0%-34.7%, p=0.003). (4) The prevalence of CHR-P state is low in the general population and ten times higher in clinical samples. The CHR-P prevalence increased with higher proportion of females in the general population and with a younger in clinical samples. The CHR-P state may be unrecognised in routine clinical practice. These findings can refine detection and preventive strategies.

  1. Page 2, Introduction:Please briefly present the epidemiology of psychosis, psychosis in psychiatric disorders and comorbidity, importance of CHR-P state detection, methods to assess CHR-P and its variability, CHR-P as an important risk biomarker, among others to lead to the importance of this study.

RESPONSE: We have expanded on some of these points in the main text and supplementary as follows:

“Psychotic disorders are frequent in the general population. The pooled incidence of psychotic disorders is 26·6 per 100 000 person-years16

“The characterization and designation of a CHR-P state is essential to guide the subsequent preventive interventions1,2 which have the potential to improve clinical outcomes3,4. Briefly, CHR-P individuals present with subtle attenuated psychotic symptoms lasting on average for more than 1 year5, frequently in association with comorbid conditions6, impairments in cognition and social cognition7. Because of these issues their quality of life may be impaired8. Brain changes have also been observed although consistent replication and clinical translations are still not fully determined5.

“CAARMS and SIPS5,17 deliver comparable prevalence of CHR-P cases, likely based on their excellent and comparable psychometric performance to discriminate those at risk or not18. Although the SIPS has shown a relatively higher sensitivity for the prediction of psychosis than the CAARMS19, this difference did not influence the prevalence of cases identified. Attenuated Psychosis Syndrome diagnosis has recently been introduced to DSM-5 and is associated with comparable prognostic accuracy6

  1. Page 6, Figure 1:The diamond is missing. The sample sizes are preferably presented in the forest plot. Please present short description in the caption.

RESPONSE:

We have added the samples sizes to the forest plot and included a diamond:

Caption: Random effect meta-analysis of the prevalence CHR-P individuals in the general population. 1.7% (95% CI=1.0-2.9%) of individuals in the general population were found to be at CHR-P.

  1. Page 7, Figure 2:The sample sizes are preferably presented in the forest plot. Please present short description in the caption.

RESPONSE:  

We have added the sample sizes

Caption: Random effect meta-analysis of the prevalence CHR-P individuals in clinical samples. 19.2% (95% CI=12.9-27.7%) of individuals in the clinical samples were found to be at CHR-P.

  1. Pages 8,9, Figure 3:Please present short description in the caption.

RESPONSE:

We have added a short description

Caption Meta-regression of the influence of sex (% female) on the prevalence of CHR-P in the General Population.  A significant increase in the prevalence of CHR-P individuals with an increasing proportion of females (β=0.045, p=0.041) was found in the general population

Caption Meta-regression of the influence of age on the prevalence of CHR-P in clinical samples.

A significant decrease in the prevalence of CHR-P individuals with an increasing age (β=-0.160, p<0.001) was found in clinical samples.

  1. Pages 9-12, Discussion:The limitation is well discussed. Please describe potentials in the study, the ultimate goal, research or knowledge needed to achieve, the future research direction, and the biggest challenge in this goal, among others.

RESPONSE:

We have now addressed these points in response to reviewer 1 as follows:

“Our findings confirm that community outreach and recruitment of CHR-P cases from the general population should only be considered if adequate risk enrichment strategies can be implemented via pre-screening assessment9, in particular via digital strategies10”.

“There are several empirical implications of this study for clinical practice and research (e.g., recruitment into studies). The main implication of this study is to have provided the first meta-analytic prevalence of the CHR-P state in the general population and in clinical samples. The general population estimate can be extremely useful to compute population-level epidemiological measures that can quantify the impact of preventive approaches (e.g. the population attributable fraction, PAF). Furthermore, this study demonstrates that within clinical samples, about five young people should be tested to identify at least one meeting the CHR-P criteria. This picture may be moderated by the use of pre-screening instruments, which led to about 5.6% false negatives, albeit only a few studies reported these data. CHR-P individuals accessing clinical services are typically complex, present with comorbid mental disorders, suicidal ideation and self-harm, as well as impairments in work/educational functioning, social functioning and quality of life5. Pre-screening instruments may have more difficulties in discriminating between these intertwined mental health problems (eDiscussion). Therefore, in clinical samples, the use of CHR-P instruments per se, without pre-screening instruments, may be sufficient. The high prevalence of CHR-P “hiding in plain sight” among adolescents and young adults in clinical services suggests that even though individuals are receiving care, their CHR-P symptoms may frequently be going unrecognised in clinics who are not specifically evaluating for them and that the opportunity to monitor these individuals and identify transitions to frank psychosis as soon as possible may be being missed. Interestingly, this finding of unrecognised CHR-P cases in clinical services may also explain the putative prognostic value of non-psychotic mental disorders for later schizophrenia11-13, raising the possibility that psychosis onset from non-psychotic disorders may be associated with undetected comorbid CHR-P. Lastly, the high prevalence of CHR-P in clinical services may offer an untapped resource for the identification of potential research participants. Since obtaining sufficient sample sizes for clinical trials within available resources is one of the principal obstacles to the development of new interventions specifically for CHR-P individuals, the possibility of improved detection of these CHR-P patients could speed the development of needed preventive interventions.”

Additional implications of the current study are appended below:

“Policy makers should make sure these services are available, particularly for help-seeking individuals under clinical care in mental health settings”

“The observed international prevalence of CHR-P features, although awaiting further replication in larger population-level studies, holds clinical value to inform the widescale implementation of specialised services to take care of individuals with emerging mental disorders14”.

  1. The manuscript contains three figures, one table and 56 references. The novelty of the content is fairly good; there will be relatively potential impact of the manuscript in the relevant field of research; the standard of English is good;  the study design and methodology are appropriate. The manuscript carries important value presenting the prevalence of CHR-P and need for fine assessment of CHR-P state for routine clinical practice.

RESPONSE:

Thank you for the feedback

Reviewer 3 Report

In the present meta-analysis, the authors aimed to systematically review the prevalence of the CHR-P individuals (clinical high-risk for psychosis) in the general population and in the clinical samples. They wanted also to address the impact of using pre-screening strategies in order to provide additional scientific knowledge of the prevalence of the CHR-P in the general population to facilitate its detection and prevention.

The manuscript is very well represented, it is clear and every single step of the procedure followed by the authors has been efficiently explained and illustrated. I believe that the present meta-analysis was properly designed. I have only some minor comments/suggestions to improve the impact of the manuscript:

  • In my opinion the abstract is too schematic. Although I appreciate concise abstracts, I believe that this abstract could be improved to highlight the main results obtained and their clinical relevance.
  • Which statistical analysis was used? If I am not mistaken, the name of the test used in the methods was not given. Authors simply describe that they used two-sided statistical tests and they reported the significance level used. Please specify this information in the main text, in the methods section.
  • Figures 1 and 2 are of low resolution. I suggest authors to improve the quality of both figures.

Author Response

Thank you for the useful feedback. Please find attached the response to reviewers

REVIEWER 3

  1. In the present meta-analysis, the authors aimed to systematically review the prevalence of the CHR-P individuals (clinical high-risk for psychosis) in the general population and in the clinical samples. They wanted also to address the impact of using pre-screening strategies in order to provide additional scientific knowledge of the prevalence of the CHR-P in the general population to facilitate its detection and prevention.

The manuscript is very well represented, it is clear and every single step of the procedure followed by the authors has been efficiently explained and illustrated. I believe that the present meta-analysis was properly designed. I have only some minor comments/suggestions to improve the impact of the manuscript:

RESPONSE:

Thank you for the positive feedback.

  1. In my opinion the abstract is too schematic. Although I appreciate concise abstracts, I believe that this abstract could be improved to highlight the main results obtained and their clinical relevance.

RESPONSE:

We have expanded the introduction and the discussion of the abstract to make it more narrative and less schematic:

Abstract: (1) The consistency and magnitude of the prevalence of Clinical High-risk for psychosis (CHR-P) individuals are undetermined, limiting efficient detection of cases. Our aim was to evaluate the prevalence of CHR-P individuals systematically assessed in the general population or clinical samples (2) PRISMA/MOOSE-compliant (PROSPERO: CRD42020168672) meta-analysis of multiple databases until 21/01/21. Random-effects model meta-analysis, heterogeneity analysis, publication bias and quality assessment, sensitivity analysis—according to the gold-standard CHR-P and pre-screening instruments—leave-one-study-out analyses and meta-regressions were conducted (3) 35 studies were included, with 37,135 individuals tested and 1,554 CHR-P individuals identified (median age=19.3 years, IQR=15.8-22.1; 52.2% females, IQR=38.7-64.4). In the general population (k=13, n=26,835 individuals evaluated), the prevalence of CHR-P state was 1.7% (95% CI=1.0-2.9%). In clinical samples (k=22, n=10,300 individuals evaluated), the prevalence of the CHR-P state was 19.2% (95% CI=12.9-27.7%). Using a pre-screening instrument was associated with false negatives (5.6%, 95% CI=2.2-13.3%) and a lower CHR-P prevalence (11.5%, 95% CI=6.2%-20.5%) compared to using CHR-P instruments only (28.5%, 95% CI=23.0%-34.7%, p=0.003). (4) The prevalence of CHR-P state is low in the general population and ten times higher in clinical samples. The CHR-P prevalence increased with higher proportion of females in the general population and with a younger in clinical samples. The CHR-P state may be unrecognised in routine clinical practice. These findings can refine detection and preventive strategies.

  1. Which statistical analysis was used? If I am not mistaken, the name of the test used in the methods was not given. Authors simply describe that they used two-sided statistical tests and they reported the significance level used. Please specify this information in the main text, in the methods section.

RESPONSE:

We used proportions as effect size and random-effects model for the meta-analysis Q statistic and the I² index for the sub-analyses and Egger’s test for the publication bias analyses. The analyses were carried out with the Meta and Metaprop packages of Stata statistical software version 16 (StataCorp) [31] and the Comprehensive Meta-Analysis Version 3 software [32]. We have now clarified this as follows:

Strategy for data synthesis and effect measures

We estimated the prevalence of CHR-P individuals systematically assessed in the general population and in clinical samples as proportions (95% CI, primary effect size). We used a random-effects model as heterogeneity was expected to be high in the studies included. Heterogeneity among study point estimates was assessed using the Q statistic and the I² index. The presence of publication bias in the results was assessed by Egger’s test20 and by visually inspecting the funnel plots, which are provided in the supplementary text. The use of the “trim and fill” method was planned to correct the effects of any publication bias detected.

We carried out a sensitivity analysis to compare a) type of CHR-P interview: studies using the SIPS vs. studies using the CAARMS and b) type of assessment: studies using only the gold-standard CHR-P instrument vs. those using first a pre-screening instrument and then the gold-standard CHR-P instrument for those individuals testing positive at the pre-screening test. Other sensitivity analyses compared c) studied conducted in school/colleges vs. other studies within the general population group and d) forensic samples vs. other samples within the clinical samples group. For comprehensiveness, we further evaluated the meta-analytical prevalence of CHR-P individuals testing negative on the pre-screening instruments but later testing positive on the CHR-P assessment (i.e., “false negatives” at the pre-screening assessment). Additional leave-one study out sensitivity analyses evaluated the stability of the meta-analytic findings when each study was removed at a time. We used meta-regression techniques to evaluate the impact of the following predictors of interest: a) age, b) sex, c) quality of the study, and d) study continent. We used two-sided statistical tests and with a significance level of alpha=0.05. The analyses were carried out with the Meta and Metaprop packages of Stata statistical software version 16 (StataCorp)21 and the Comprehensive Meta-Analysis Version 3 software22.

  1. Figures 1 and 2 are of low resolution. I suggest authors to improve the quality of both figures.

RESPONSE:

We have improved the quality of both figures as follows:

Figure 1:

Figure 2:

Round 2

Reviewer 2 Report

18 October 2021

Manuscript ID: brainsci-1443994

Type: Review

Title: “Prevalence of Individuals at Clinical High-risk of Psychosis in the General Population and Clinical Samples: Systematic Review and Meta-analysis” by Salazar de Pablo G et al., submitted to Brain Sciences

Dear Authors,

Efficient detection of clinical high-risk for psychosis (CHR-P) state is the most important step for the prevention of psychosis; however, little is known about the prevalence of CHR-P. The authors conducted a meta-analysis of the prevalence of CHR-P individuals in the general population and clinical samples. The study concluded that the prevalence of CHR-P is ten times higher in clinical samples than general population and that CHR-P state may remain unrecognized in routine clinical practice.

The manuscript contains three figures, one table and 56 references. The authors addressed the response properly. The manuscript is revised accordingly and thus, the quality is substantially improved. The manuscript carries important value presenting the prevalence of CHR-P and need for fine assessment of CHR-P state for routine clinical practice. 

Best regards,